# The Effect of Scapular Fixation on Scapular and Humeral Head Movements during Glenohumeral Axial Distraction Mobilization

**DOI:** 10.3390/medicina58030454

**Published:** 2022-03-21

**Authors:** Carlos López-de-Celis, Elena Estébanez-de-Miguel, Albert Pérez-Bellmunt, Santos Caudevilla-Polo, Vanessa González-Rueda, Elena Bueno-Gracia

**Affiliations:** 1Faculty of Medicine and Health Sciences, Universitat Internacional de Catalunya, 08195 Sant Cugat del Vallès, Spain; carlesldc@uic.es (C.L.-d.-C.); aperez@uic.es (A.P.-B.); vgonzalez@uic.es (V.G.-R.); 2ACTIUM Functional Anatomy Group, Universitat Internacional de Catalunya, 08195 Sant Cugat del Vallès, Spain; 3Institut Universitari per a la Recerca a l’Atenció Primària de Salut Jordi Gol i Gurina (IDIAPJGol), 08007 Barcelona, Spain; 4Faculty of Health Sciences, University of Zaragoza, 50011 Zaragoza, Spain; scp@unizar.es (S.C.-P.); ebueno@unizar.es (E.B.-G.)

**Keywords:** glenohumeral joint, manual therapy, mobilization, ultrasound

## Abstract

*Background and Objectives:* Glenohumeral axial distraction mobilization (GADM) is a usual mobilization technique for patients with shoulder dysfunctions. The effect of scapular fixation on the movement of the scapula and the humeral head during GADM is unknown. To analyze the caudal movement of the humeral head and the rotatory movement of the scapula when applying three different intensities of GADM force with or without scapular fixation. *Materials and Methods:* Fifteen healthy subjects (mean age 28 ± 9 years; 73.3% male) participated in the study (twenty-eight upper limbs). Low-, medium- and high-force GADM in open-packed position were applied in scapular fixation and non-fixation conditions. The caudal movement of humeral head was evaluated by ultrasound measurements. The scapular rotatory movement was assessed with a universal goniometer. The magnitude of force applied during GADM and the region (glenohumeral joint, shoulder girdle, neck or nowhere) where subjects felt the effect of GADM mobilization were also recorded. *Results:* A greater caudal movement of the humeral head was observed in the non-scapular fixation condition at the three grades of GADM (*p* < 0.008). The rotatory movement of the scapula in the scapular fixation condition was practically insignificant (0.05–0.75°). The high-force GADM rotated scapula 18.6° in non-scapular fixation condition. Subjects reported a greater feeling of effect of the techniques in the glenohumeral joint with scapular fixation compared with non-scapular fixation. *Conclusions:* The caudal movement of the humeral head and the scapular movement were significantly greater in non-scapular fixation condition than in scapular fixation condition for the three magnitudes of GADM force.

## 1. Introduction

Shoulder pain is a common pathology in primary care medical practice and orthopaedic surgery with an annual incidence of 17.3 per 1000 in subjects between 45 and 64 years old, and 12.8 per 1000 in subjects between 65 and 74 years old [1]. In addition to pain, restriction of movement is one of the most frequent consequences of shoulder pathologies [2,3]. Clinical practice guidelines recommend joint mobilization procedures primarily directed to the glenohumeral joint to relieve pain, increase shoulder mobility and improve physical function in patients with adhesive capsulitis [3].

Translational mobilizations are widely accepted for the treatment of shoulder diseases [4,5,6,7,8,9]. The humeral head performs a linear translational movement with respect to the articular surface of the scapula in this type of mobilization technique [10,11]. Glenohumeral axial distraction mobilization (GADM) has been shown to be effective to decrease pain and increase mobility in many shoulder dysfunctions [12,13,14,15,16,17,18,19]. The GADM has been shown to produce a caudal humeral head gliding and an increase in the subacromial space [7,20,21].

In clinical practice, scapular fixation is often performed to prevent movement of the scapula during the application of GADM. The scapular fixation is supposed to reduce the transmission of undesirable forces to other structures in the scapulothoracic and cervical region, which could generate harmful stress [22,23,24]. It has been suggested that the scapular fixation could focus the effect of the mobilization technique on the glenohumeral joint, increasing the caudal movement of the humeral head and avoiding scapular rotatory movement. However, how the scapular fixation can affect the movement of the scapula and the humeral head during GADM is unknown.

The purposes of this study were (1) to measure and compare the caudal movement of the humeral head and the scapular rotatory movement when applying three different magnitudes of forces (low, medium and high) during GADM, with and without scapular fixation and (2) to determine the regions (glenohumeral joint, shoulder girdle, neck or nowhere) where subjects feel the effect of the three GADM forces, in scapular fixation and non-fixation condition.

## 2. Materials and Methods

A cross-sectional study was carried out. A repeated-measures design was used to caudal movement of humeral head and analyse scapular rotatory movement during three magnitudes of GADM force (low, medium, high) across two conditions (scapular fixation or scapular non-fixation).

The main study variables were the caudal movement of the humeral head and the scapular rotatory movement. The secondary variable was the region (glenohumeral joint, shoulder girdle, neck or nowhere) where subjects felt the effect of the three magnitudes of GADM force.

This study was conducted with ethical approval from the institutional ethics committee of the Universitat Internacional de Catalunya (CBAS-2021-15). The procedures followed were in accordance with the Declaration of Helsinki 1975, revised Fortaleza 2013.

Sample size calculation was carried out based on the magnitude of the difference found in the variable caudal movement of the humeral head during distraction mobilizations (d = 1.58), and the standard deviation (2.12) of this variable based on previous pilot study (*n* = 10); the level of significance of 0.05, a power of 0.8 and no follow-up loss rate. According to these parameters, the sample size was constituted by 28 upper-limbs.

The sample was recruited from volunteers of the Universitat Internacional de Catalunya. The inclusion criteria were as follows: subjects over 18 years old who signed the informed consent. The exclusion criteria were as follows: (1) pain in the shoulder region, (2) history of orthopaedic injuries in the shoulder region, (3) medical diagnosis of connective tissue involvement. Fifteen volunteers (30 upper-limbs) who fulfilled inclusion and exclusion criteria were recruited. Two upper-limbs were excluded for presenting pain.

Testing was performed in a single session. Sociodemographic characteristics were registered at the beginning of the session: age, sex, arm dominance, height, weight and body mass index.

All GADM techniques were performed by a single physical therapist who had more than 15 years of clinical experience. A second physical therapist, with more than 5 years of musculoskeletal ultrasound imaging experience, completed all ultrasound imaging. A third physical therapist measured scapular rotatory movement with a universal goniometer and registered the magnitude of force applied during low-, medium- and high-force GADM and the region where the participant felt the effect of joint mobilization.

Initially, all measurements were performed in scapular fixation condition. Subjects were positioned in supine, and a belt under the armpit in a cranial and medial direction was placed for scapular fixation. A small padding around the belt was used to prevent skin injury and a joint distraction cuff was placed around the distal part of the arm. For the GADM technique, the mobilizing physical therapist placed a mobilization belt around her pelvis. This mobilization belt was attached to the distraction cuff and a dynamometer (475055 Digital Force Gauge; Extech, Boston, MA, USA) was placed between them to measure the magnitude of applied force (low-, medium- and high-force GADM).

The physical therapist who applied the GADM was blinded both to the magnitude of force exerted and ultrasound images. The third physical therapist registered force data (Figure 1).

The glenohumeral joint was placed in its open-packed position (55° abduction and 30° horizontal adduction of arm) [10]. Subjects were instructed to keep their arm relaxed. A 40 mm linear transducer of a portable US machine (Samsung—digital ultrasound HS30) was placed vertically over the lateral aspect of the glenohumeral joint in the suprahumeral joint space. The transducer was translated medially and laterally until the acromion process and the superior aspect of the humeral head were visible in the ultrasound imaging.

The mobilizing therapist pulled axially with three different magnitudes of GADM force according to Kaltenborn’s grades of joint mobilization with glenohumeral joint in open-packed position (55° abduction and 30° horizontal adduction of arm) [10]. An ultrasound image and scapular position was taken as baseline ultrasound images, scapular rotatory movement, the region where the subject felt the technique and the associated magnitude of force applied were recorded in the tree magnitudes of GADM. The low-force GADM was when the physical therapist verbally indicated that the slack of the joint was taken up. The medium-force GADM was when a marked resistance (the “first stop”) was first felt, and the high-force GADM was when there was the maximal resistance of the tissues. This procedure was applied in the same sequence and repeated twice with 30 s rest between repetitions.

Then, scapular non-fixation condition was performed removing the armpit belt for scapular fixation. The subject maintained a supine position with glenohumeral joint in open-packed position, but the armpit belt for scapular fixation was removed. An ultrasound image and scapular position was taken as baseline measurement in scapular non-fixation condition. The magnitude of force applied during the low, medium and high GADM in scapular non-fixation condition reproduced the mean values recorded during scapular fixation condition. The mobilizing therapist pulled until the third physical therapist verbally indicated to stop because the mean value had been reached. Ultrasound images, scapular rotatory movement and the region where the subject felt the effect of mobilization were also recorded in each magnitude of GADM force twice.

The caudal movement of the humeral head was measured by a single examiner blinded to the scapular fixation condition and the magnitude of force applied in each analysed image. Ultrasound images were exported as jpg file and ImageJ software (https://imagej.nih.gov/ij/docs/guide/ accessed on 30 July 2020) was used for all measurements. On each image, a horizontal line was drawn at the superior acromion level, a second horizontal line was drawn tangent to the superior humeral head and the distance between the two horizontal lines was measured as caudal movement of humeral head. The amount of caudal movement of the humeral head for each magnitude of force mobilization was calculated by subtracting the position of the humeral head after a mobilization from the position of the humeral head at rest (Figure 2). The mean of the distance registered in the two trials was used for statistical analysis.

To determine intra-observer reliability of the ultrasound images measurements, two assessments where made in 10 subjects with the same characteristics of the study sample, 48 h prior to the present study. The intraclass correlation coefficient (two-way mixed-effect model) (ICC3,1), standard error of measurement (SEM), and the minimal detectable change at the 95% confidence level (MDC95%) for ultrasound measurements of the caudal movement of humeral head are displayed in Table 1.

The scapular movement was measured with a universal goniometer. The center of the goniometer was placed on the middle of the glenohumeral joint, on the dorsal side. The stationary goniometer arm was placed along the diaphysis of the humerus and the moving goniometer arm was placed in alignment with the lateral border of the scapula. Starting from this position, the changes in the different magnitudes of force applied were being measured, following the same references. The mean of the scapular rotation position recorded in the two trials for each experimental condition and magnitude of force applied was used for statistical analysis.

Subjects were being asked in which region have felt the effect of GADM. They could identify the region with the help of a human body diagram to indicate that mobilization has been felt in the glenohumeral joint, shoulder girdle, neck or nowhere. Subjects could indicate only one region.

IBM SPSS Statistics for Windows, Version 20.0. (Armonk, NY, USA: IBM Corp) was used for all statistical analyses. Descriptive statistics (mean and standard deviations, or number and percentage) were calculated to describe the demographic characteristics of sample.

Two-way repeated measures ANOVA was used to analyse the differences in the caudal movement of humeral head and scapular movement with the magnitude of GADM force and fixation/no fixation condition as factors. If ANOVA was found to be significant, Bonferroni-adjusted post hoc tests were used to assess pairwise comparisons. The Fisher’s exact test was used to investigate the differences in the identification of the region where subjects felt the effect of GADM between fixation and non-fixation conditions in each magnitude of GADM force. The effect size was calculated to estimate the magnitude of the difference between two conditions, on the main variables, with Cohen coefficient (d). Cohen coefficients were interpreted as follows: large effect sizes, d > 0.8; moderate effect sizes, d = 0.5–0.79; and small effect sizes, d = 0.2–0.49 [25].

## 3. Results

Twenty-eight upper-limbs from fifteen voluntary subjects (11 male, 4 female) were examined in both fixation and non-fixation conditions. The demographic characteristics of the sample are shown in Table 2.

Table 3 provides data of the magnitude of the applied force, the caudal movement of the humeral head and the scapular movement in scapular fixation and non-fixation conditions for each magnitude of GADM force, the mean difference and 95% CI, the effect sizes and the interaction effects. Two-way repeated measures ANOVA showed a significant interaction between the two factors in the caudal movement of the humeral head (F = 5.262 *p* = 0.008) and the scapular movement variables (F = 1966.56 *p* < 0.001). Bonferroni post hoc tests revealed significant differences in the caudal movement of the humeral head and the scapular movement between scapular fixation and non-fixation conditions. The caudal movement of humeral head and scapular movement were greater when the three magnitudes of GADM forces were applied without scapular fixation. The greatest mean difference in caudal movement of humeral head was showed during medium-force GADM, and the greatest mean difference in scapular movement was showed during high-force GADM.

In relation to the magnitude of GADM force, the caudal movement of the humeral head increased significantly with greater magnitudes of force in both conditions (*p* < 0.001). For scapular movement, similar results were shown when the three magnitudes of force were applied in scapular non-fixation condition (*p* < 0.001). However, in the fixation condition, the scapular movement was significantly greater during medium- (*p* < 0.001) and high-force (*p* < 0.001) GADM compared to low-force GADM, but no significant differences were found between medium- and high-force GADM (*p* = 0.859).

Table 4 provides the data for the region in which the subjects felt the effect of the different magnitudes of the GADM technique according to the force applied during the two trials. Differences between the scapular fixation and non-scapular fixation were statistically significant *p* < 0.05 according to Fisher’s exact test.

## 4. Discussion

To our knowledge, this is the first study to analyze the effect of scapular fixation on the caudal movement of the humeral head and the rotatory movement of the scapular when applying three different magnitudes of forces during GADM. The results of the present study showed that the caudal movement of the humeral head and the scapular movement were significant greater in non-scapular fixation condition than in scapular fixation condition for the three magnitudes of GADM force.

Ultrasound measurements showed that the caudal movement of the humeral head in each magnitude of force during the GADM was greater in the condition in the non-fixation condition. The amount of caudal movement of the humeral head during the GADM was similar to that found in the study of Witt and Talbott [20], except for the high-force application. During high-force application, a greater caudal movement of the humerus head was observed both in the scapular fixation condition (a difference of 1.78 mm was found) and in the scapular non-scapular fixation condition (difference of 2.56 mm) in the Witt and Talbott [20] study. As it was pointed out, these differences may be related to the greater force applied at high-force application in the present study.

The fact that the caudal movement of the humeral head was higher during the application of the GADM in the non-scapular fixation condition could be related to the tilting of the scapula observed during the application of the technique. The scapular movement produced an increase of 18.6° in the shoulder abduction at high-force application, which was not observed during the application of the GADM with scapular fixation. The scapular rotatory movement during the application of the GADM was already described by [26]. In their study, an increase of 8° in shoulder abduction was reported during the application of 4 kg of axial traction. This amount of force is similar to the force applied at medium force in the present study, where an increase of 6.8 in shoulder abduction was found. Biomechanically, a caudal gliding of the humeral head has been described to occur during the shoulder abduction [10]. The scapular rotatory movement could support an increase in the distance between the acromion and the humerus in the condition of non-scapular fixation. This movement could generate a non-real gain in the caudal gliding.

Subjects reported to feel the effect of medium- and high-force GADM on glenohumeral joint when the scapula was fixated. However, the effect was felt on the shoulder girdle or neck when the scapula was not fixated. This could be relevant in clinical practice. Following Kaltenborn’s indications [10], low force and medium force in traction techniques are used for pain relief, so it is interesting to avoid scapular rotatory movement. High force is used to achieve joint gliding and tissue elongation [27,28,29]. For this reason, focusing the force on the treated region minimizes the stress dispersion to other anatomical structures. The present study did not measure which structures experienced an increase in stress or the amount of stress during the GADM. Therefore, by the force on the treated region, the dispersion of stress to other structures is minimized. Studies similar to those carried out by Estébanez-de-Miguel [30,31,32] in the hip region would be needed to know the amount of stress received by the different ones and its relationship with the force applied during mobilization.

Therefore, in clinical practice, when shoulder patients are treated with GADM, it would be important to fix the scapula to cause a specific movement of the humeral head and avoid unnecessary stress on other anatomical structures. However, if the scapula has fractures that compromise its stability or require surgical intervention, this technique may not be appropriate [33].

The magnitudes of GADM force used in the present study were higher than in the study of Witt and Talbott [20], where a similar glenohumeral position was set. In the study of Witt and Talbott [20], the force intensity at grades 1 and 2 was lower than in the present study, with values of 37.4 ± 5.3 N for grade 1 and 91.2 ± 11.1 N for grade 2. while in the present study, the values of applied force were 16.21 ± 5.1 N and 46.71 ± 12.4 N, respectively. However, for grade 3, a higher force was applied in the present study, with a difference of 60.9 N.

The difference in high-force application technique could be attributed to subjective differences between evaluators, but also to the direction of the mobilization in each study.

Regarding the direction of the applied force, in the study of Witt and Talbott [20], the direction of the force was caudal, while an axial force was applied in the present study. Garwood et al. [26] reported a force application of 4 kg during axial traction of the shoulder, similar to grade 2 of the present study. The higher force applied in the present study compared to the study of Garwood et al. [26] could be related to the subjects’ ability to remain relaxed during the application of the GADM. Although subjects were instructed to remain relaxed during the technique application, a defensive response to traction could occur and made the physical therapist increase the intensity of the technique. The reliability of detection of the grades of movement has already been evaluated in other studies, showing a good or an excellent intra-observer [30,34,35,36,37,38,39]. Ultrasound measurements have been shown to be a reliable tool to evaluate inferior gliding to humeral head [35,36]. In the present study, an excellent intra-observer reliability (ICCs over 0.99), exceeding the SEM in all force magnitudes, was observed.

### Limitations

This study presents limitations. First, no randomization of techniques in scapular fixation or non-scapular fixation conditions was performed due to methodological reasons. Therefore, the effect of repetition on shoulder tissues could have influenced the measurements and results should be taken with caution. Second, because the study was performed on asymptomatic subjects, the results obtained in the present study cannot be extrapolated to patients with shoulder disorders. Third, even though the subjects were given the order to keep their arm relaxed during the application of the GADM, potential muscle guarding was not controlled in the study, so this could have altered some results. Finally, the results obtained are based on a single session of experimental mobilisation, so we do not know if this result can be the same in a clinical context where more time and more treatment sessions are applied.

## 5. Conclusions

The GADM in non-scapular fixation condition produced greater caudal movement of the humeral head compared to the scapular fixation condition. The scapular position did not change during GADM with scapular fixation condition. The perception of the greatest effect on the glenohumeral joint was with the scapular fixation condition. In clinical practice, it would be important to fix the scapula to cause a specific movement of the humeral head and avoid unnecessary stress on other anatomical structures.

## Figures and Tables

**Figure 1 medicina-58-00454-f001:**
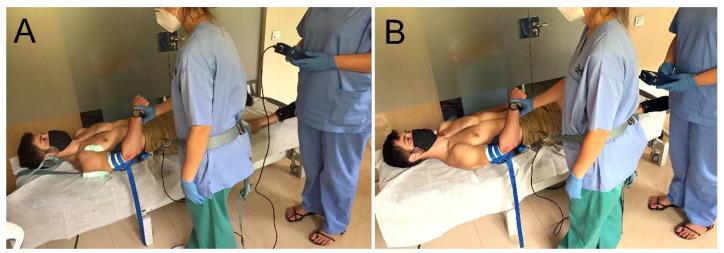
Experimental set-up. Axial traction glenohumeral joint mobilization technique. (**A**): Scapular fixation condition; (**B**): Non-scapular fixation condition.

**Figure 2 medicina-58-00454-f002:**
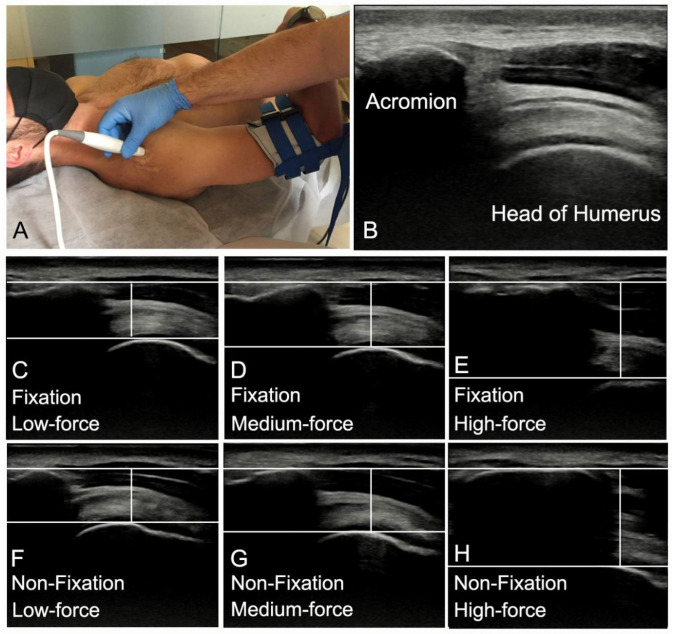
(**A**) Transducer position. Ultrasound image at rest with the acromion and humeral landmarks identified. (**B**): Ultrasound image of humeral head position during at rest as baseline. Scapular fixation ultrasound image of humeral head position on low-force GADM (**C**), medium-force GADM (**D**) and high-force GADM (**E**). Non scapular fixation ultrasound image of humeral head position on low-force GADM (**F**), medium-force GADM (**G**) and high-force GADM (**H**).

**Table 1 medicina-58-00454-t001:** Outcomes of ultrasound measurements of caudal movement of humeral head during three magnitudes of GADM force.

Magnitude of Force	ICC3, 1 (95% CI)	SEM	MDD95
Baseline	0.996 (0.986–0.999)	0.01 mm	0.02
Low-force GADM	0.993 (0.972–0.998)	0.02 mm	0.05
Medium-force GADM	0.996 (0.972–0.998)	0.01 mm	0.02
High-force GADM	0.997 (0.990–0.999)	0.01 mm	0.02

Abbreviations: ICC3, 1: Intraclass Correlation Coefficient, 95% CI: 95% Confidence Level, SEM: Standard Error of Measurement, MDC95: Minimum Detectable Change at the 95% confidence level, GADM: glenohumeral axial distraction mobilization.

**Table 2 medicina-58-00454-t002:** Subjects demographic characteristics.

	Mean ± SD or N (%)
Age (year)	28.67 ± 9.31
Gender	
Men	11 (73.3%)
Women	4 (26.7%)
Dominance	
Right	13 (86.7%)
Left	2 (13.3%)
Height (cm)	175 ± 8.28
Weight (kg)	71.80 ± 11.80
BMI (kg/m^2^)	23.40 ± 2.47

Abbreviations: SD, standard deviation; cm, centimeter; kg, kilogram; BMI, body mass index; *n*, number.

**Table 3 medicina-58-00454-t003:** Outcomes of the magnitude of force applied with GADM, with and without scapular fixation in caudal movement of the humeral head and scapular movement.

Variable	Magnitude of GADM Force	Scapular Fixation	Non-Scapular Fixation	Mean Difference (95%CI)	Effect Size	*p* Value
Caudal movement of the humeral head	Low-force (16.21 ± 5.10 N)	0.74 ± 0.55 mm	1.21 ± 0.67 mm	0.47 mm (0.11, 0.83) *p* = 0.012	0.78	F = 5.262 *p* = 0.008
	Medium-force (46.71 ± 12.39 N)	1.62 ± 0.82 mm	3.04 ± 1.45 mm	1.42 mm (0.89, 1.94) *p* < 0.001	1.21	
	High-force (200.98 ± 51.19 N)	5.38 ± 1.95 mm	6.15 ± 2.39 mm	0.77 mm (1.02, 1.45) *p* = 0.026	0.35	
Scapular movement	Low-force (16.21 ± 5.10 N)	55.05 ± 0.12°	57.06 ± 57°	2.01° (1.78, 2.23) *p* < 0.001	4.88	F = 1966.56 *p* < 0.001
	Medium-force (46.71 ± 12.39 N)	55.75 ± 0.58°	62.58 ± 78°	6.84° (6.49, 7.19) *p* < 0.001	9.94	
	High-force (200.98 ± 51.19 N)	55.91 ± 0.12°	74.55 ± 1.37°	18.63° (18.80, 19.18) *p* < 0.001	19.17	

Abbreviations: GADM, glenohumeral axial distraction mobilization; N, Newtons; mm, millimeters; º, grades.

**Table 4 medicina-58-00454-t004:** Region where subjects felt the effect of the techniques.

	Low-Force		Medium-Force		High-Force	
	T1 N (%)	T2 N (%)	T1 N (%)	T2 N (%)	T1 N (%)	T2 N (%)
Scapular Fixation						
Nowhere	22 (78.6%)	23 (82.1%)	2 (7.1%)	1 (3.6%)	0 (0%)	0 (0%)
Glenohumeral joint	6 (21.4%)	5 (17.9%)	26 (92.9%)	27 (96.4%)	24 (85.7%)	24 (85.7%)
Shoulder girdle	0 (0%)	0 (0%)	0 (0%)	0 (0%)	4 (14.3%)	4 (14.3%)
Neck	0 (0%)	0 (0%)	0 (0%)	0 (0%)	0 (0%)	0 (0%)
Non-scapular Fixation						
Nowhere	22 (78.6%)	21 (75%)	17 (60.7%)	17 (60.7%)	4 (14.3%)	4 (14.3%)
Glenohumeral joint	0 (0%)	0 (0%)	2 (7.1%)	1 (3.6%)	0 (0%)	0 (0%)
Shoulder girdle	5 (17.9%)	2 (7.1%)	6 (21.4%)	6 (21.4%)	18 (64.3 %)	16 (57.1%)
Neck	1 (3.6%)	5 (17.9%)	3 (10.7%)	4 (14.3%)	6 (21.4%)	8 (28.6%)
*p* value *	<0.002	<0.001	<0.001	<0.003	<0.001	<0.001

Abbreviations: T1: Trial 1, T2: Trial 2, *n*: number, %: percentage; *: Fisher’s exact test.

## Data Availability

The data presented in this study are available on request from the corresponding author.

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
