# Peer review of "The Effect of Scapular Fixation on Scapular and Humeral Head Movements during Glenohumeral Axial Distraction Mobilization"

_medicina, 2022, doi:10.3390/medicina58030454_

Round 1

Reviewer 1 Report

How this research design was "repeated measures" as is stated in line 67 since, in line 90, the authors claim that the testing was performed in a single session?

Perhaps the authors should consider if their research is based on group comparison.

Line 84 What does "volunteers from the X" mean?
Which were the exclusion criteria?
Didi the authors considered distinct types of "orthopaedic injuries in 86 the shoulder region"? The pathologies can be varied.
How was the diagnosis of connective tissue involvement performed? Only in the limitation section is specified the fact that the subjects were asymptomatic. This should also be mentioned in the participants' section.

The second physiotherapist who applied GADM, could not be totally blinded, although she could not see the dynamometer measurement, as an experienced physiotherapist, she might appreciate from clinical practice the force applied, as the authors state also in lines 124-128. Please revise the methodology of the research and reconsider statements. Or describe in parallel the protocols for each group.

In the limitation section, the authors should also emphasise that the research was based on only one session of mobilization, and emphasise that was made on healthy subjects, therefore the results can not be extrapolated to shoulder disorders.

Author Response

Thank you for your input and your time in conducting the review.

How this research design was "repeated measures" as is stated in line 67 since, in line 90, the authors claim that the testing was performed in a single session?

The test was conducted in a single session. However, four measurements were performed, the baseline and the three magnitudes of forces applied in each condition (Fixation - Non-fixation) Line 121-124. For this reason, we consider it to be a repeated measures design.

Perhaps the authors should consider if their research is based on group comparison.

We think that this is not a group comparison design, as it is the same subject in which four repeated measurements are performed with two different conditions.

Line 84 What does "volunteers from the X" mean?

Thank you for detecting the mistake. It has been corrected.  “…volunteers from the Universitat Internacional de Catalunya.”

Which were the exclusion criteria?
The exclusion criteria are explained on lines 85-89.

Didi the authors considered distinct types of "orthopaedic injuries in 86 the shoulder region"? The pathologies can be varied.

Thank you for your comment. We agree that injuries can be varied. We wanted to group them all under the term orthopaedic injuries, referring to a history of injury, trauma or known deformity. Only subjects with no history of shoulder pathology are included.

How was the diagnosis of connective tissue involvement performed? Only in the limitation section is specified the fact that the subjects were asymptomatic. This should also be mentioned in the participants' section.

Thank you for your comment. One of the exclusion criteria was that the subjects had a medical diagnosis of connective tissue involvement. Line 87. This sentence has been clarified.

The second physiotherapist who applied GADM, could not be totally blinded, although she could not see the dynamometer measurement, as an experienced physiotherapist, she might appreciate from clinical practice the force applied, as the authors state also in lines 124-128. Please revise the methodology of the research and reconsider statements. Or describe in parallel the protocols for each group.

We consider that the physiotherapist who applied the technique was blind to the application, as she did not have visual access to the applied force. Furthermore, she also did not have visual access to the ultrasound measurement. As discussed in the review, being an experienced physiotherapist, she is likely to detect the degrees of movement better.  

In lines 124-128, the description of the therapist's sensation in the different degrees of traction is pedagogically used to convey the force to be applied.

In the limitation section, the authors should also emphasise that the research was based on only one session of mobilization, and emphasise that was made on healthy subjects, therefore the results can not be extrapolated to shoulder disorders.

Thank you for your comments.

The limitation suggested by the reviewer that the study is based on a single mobilization session has been added. The second limitation you comment on is already described (line 312-314), but "to shoulder disorders" has been specified.

Reviewer 2 Report

Dear Authors

I have reviewed your paper with great interest.

My revision is:

Title: Very Good

Abstract: Very Good

Introduction and AIM: The problem and the aim are well descripting.

Marterials, Patients and methods and statistics: All good.

Results: Focus on and well described.

Discussion and Thread: effectiveness Focus ON.
Did you consider this problem before to fix the scapuka?
lateral border off set(medialization/lateralization) 15mm≤ or
≥20mm; glenoid angle ≥ 22°; angulation ≥45°; Scapular fractures with ≥15mm medialization +
≥30mm angulation; double disruption: Scapular fracture >10mm displaced on any view +
Clavicle fracture > 10mm displaced on any view; complete acromion-clavicular disruption; glenoid
fracture; proximal humeral fracture, homolateral clavicle fracture, bilateral shoulder injuries, nerve
injuries; vascular injuries, age under 16 years, hematological or oncological pathology; shoulder
bone metabolism disease, shoulder osteoarthritis; rheumatoid disease.

The assessment of outcomes in the scapula traumatology to evaluate the psychological aspect, and Shoulder Range of Motion, cite and discuss this paper:

Rollo G, Huri G, Meccariello L, Familiari F, Çetik RM, Cataldi C, Conteduca J, Giaracuni M, Bisaccia M, Longo D, Giannotti PS. Scapular body fractures: Short-term results of surgical management with extended indications. Injury. 2021 Mar;52(3):481-486. doi: 10.1016/j.injury.2020.09.006. Epub 2020 Sep 14. PMID: 32951918.

References: Well chosen but to improve

Figures and Table: Very Good.

Author Response

Thank you for your interest and your time in the review.

I have reviewed your paper with great interest.

My revision is:

Title: Very Good

Abstract: Very Good

Introduction and AIM: The problem and the aim are well descripting

Marterials, Patients and methods and statistics: All good..

Results: Focus on and well described

Discussion and Thread: effectiveness Focus ON.

Thank you for your comments

Did you consider this problem before to fix the scapuka?

lateral border off set(medialization/lateralization) 15mm≤ or ≥20mm; glenoid angle ≥ 22°; angulation ≥45°; Scapular fractures with ≥15mm medialization +≥30mm angulation; double disruption: Scapular fracture >10mm displaced on any view + Clavicle fracture > 10mm displaced on any view; complete acromion-clavicular disruption; glenoid fracture; proximal humeral fracture, homolateral clavicle fracture, bilateral shoulder injuries, nerve injuries; vascular injuries, age under 16 years, hematological or oncological pathology; shoulder bone metabolism disease, shoulder osteoarthritis; rheumatoid disease.

Thank you for your comment. Subjects with a history of shoulder pathology were previously excluded from the study. Line 86.

The assessment of outcomes in the scapula traumatology to evaluate the psychological aspect, and Shoulder Range of Motion, cite and discuss this paper:

Rollo G, Huri G, Meccariello L, Familiari F, Çetik RM, Cataldi C, Conteduca J, Giaracuni M, Bisaccia M, Longo D, Giannotti PS. Scapular body fractures: Short-term results of surgical management with extended indications. Injury. 2021 Mar;52(3):481-486. doi: 10.1016/j.injury.2020.09.006. Epub 2020 Sep 14. PMID: 32951918.

Your comment has been taken into consideration, taking into account the article you propose. Thank you.

References: Well chosen but to improve

Figures and Table: Very Good.

Thanks for your comments.

Round 2

Reviewer 1 Report

I am fine with the adjustments.